# Carbon Ion Radiobiology

**DOI:** 10.3390/cancers12103022

**Published:** 2020-10-17

**Authors:** Walter Tinganelli, Marco Durante

**Affiliations:** 1Biophysics Department, GSI Helmholtzzentrum für Schwerionenforchung, Planckstraße 1, 64291 Darmstadt, Germany; w.tinganelli@gsi.de; 2Institut für Festkörperphysik, Technische Universität Darmstadt, Hochschulstraße 8, 64289 Darmstadt, Germany

**Keywords:** carbon ions, particle therapy, radiotherapy, radiobiology, hypoxia, RBE, immunotherapy, metastasis

## Abstract

**Simple Summary:**

Radiotherapy with carbon ions has been used for over 20 years in Asia and Europe and is now planned in the USA. The physics advantages of carbon ions compared to X-rays are similar to those of protons, but their radiobiological features are quite distinct and may lead to a breakthrough in the treatment of some cancers characterized by high mortality.

**Abstract:**

Radiotherapy using accelerated charged particles is rapidly growing worldwide. About 85% of the cancer patients receiving particle therapy are irradiated with protons, which have physical advantages compared to X-rays but a similar biological response. In addition to the ballistic advantages, heavy ions present specific radiobiological features that can make them attractive for treating radioresistant, hypoxic tumors. An ideal heavy ion should have lower toxicity in the entrance channel (normal tissue) and be exquisitely effective in the target region (tumor). Carbon ions have been chosen because they represent the best combination in this direction. Normal tissue toxicities and second cancer risk are similar to those observed in conventional radiotherapy. In the target region, they have increased relative biological effectiveness and a reduced oxygen enhancement ratio compared to X-rays. Some radiobiological properties of densely ionizing carbon ions are so distinct from X-rays and protons that they can be considered as a different “drug” in oncology, and may elicit favorable responses such as an increased immune response and reduced angiogenesis and metastatic potential. The radiobiological properties of carbon ions should guide patient selection and treatment protocols to achieve optimal clinical results.

## 1. Introduction

The interest in carbon ion radiobiology derives from the use of these nuclei in cancer radiotherapy. The original idea of Robert R. Wilson, a student of Ernest Orlando Lawrence at the University of California in Berkeley (CA, USA), was to exploit the Bragg peak to improve the dose-depth distribution in radiotherapy [1], and this can be done with fast protons. Using heavier ions in cancer therapy was proposed and pursued by Cornelius A. Tobias at the Lawrence Berkeley Laboratory (LBL) [2], with the idea that heavier ions have radiobiological properties that can lead to better results than X-rays or protons. The physical properties of heavy ions and protons are similar (Figure 1) [3,4]. Heavy ions have reduced lateral scattering and longitudinal straggling compared to protons, but they present a tail of light fragments beyond the Bragg peak and, for very heavy ions, nuclear fragmentation causes a decrease of the dose in the plateau region, as is commonly observed in tests of shielding materials using very heavy ions at energies characteristic of the cosmic ray spectrum [5]. 

However, the relative biological effectiveness (RBE) increases with the radiation linear energy transfer (LET) [6,7,8,9,10], and because the LET is proportional to z^2^/β^2^ (where z is the ion effective charge and β its relative velocity), fast ions (in the entrance channel) will have a lower RBE than slow ions in the target region. Other biological effects of high-LET ions, such as the reduced dependence on fractionation and cell-cycle stage, and especially the reduced oxygen enhancement factor (OER), made them attractive for treatment of radioresistant malignancies (Figure 2) [11,12]. Patients in Berkeley were treated in the period 1975–1992 with several ions: He, N, O, C, Ne, Si, and Ar [13]. The use of very heavy ions, such as argon, was justified by the goal of overcoming hypoxia, one of the major causes of cancer radioresistance [14], which requires very high LET according to the cell experiments performed at the LBL Bevalac accelerator [8]. However, the entrance LET of heavy ions is relatively high, resulting in unacceptable patient toxicities. Carbon represents a good compromise, with an LET in the entrance channel between 11 and 13 keV/μm, and a fairly high LET on the Spread-out-Bragg-peak (SOBP) between 40-80 keV/μm (Figure 3) [15]. For this very reason, Hirohiko Tsujii and his colleagues at the National Institute of Radiological Sciences (NIRS) in Chiba, Japan (now re-named National Institute for Quantum and Radiological Science and Technology) elected to use ^12^C-ions for therapy in 1994. They have now treated over 13,000 patients with carbon ions [16,17], by far the largest cohort worldwide with these ions, and their success inspired Gerhard Kraft to start a clinical trial with C-ions at the GSI Helmholtzzentrum für Schwerionenforschung [18,19], followed by clinical projects in different European countries [20]. 

According to the Particle therapy co-operative Group (PTCOG) (https://www.ptcog.ch/) database, by the end of 2018 there were eight C-ion centers in Asia and four in Europe that have treated approximately 28,000 patients, and eight more centers are under construction or in the planning stage. Mayo Clinics in Jacksonville (FL, USA) has advanced plans for building the first C-ion therapy center in the USA [21]. This facility would mark a much-awaited return of heavy ion therapy in the USA after the end of the LBL pilot trial. Concerns with respect to the cost effectiveness and lack of FDA certification and approved reimbursement have hampered its introduction in the US so far [22]. The high investment cost is also affecting the well-established proton therapy [23], but the carbon cost is even higher, causing resilience even in the proton therapy community [24]. Yet some of the clinical results with C-ions, such as the long-term survival of chordoma patients in Germany [25] and locally advanced pancreatic cancer [26,27] and locally recurrent rectal cancer [28] in Japan, are clearly superior to both X-rays and protons [29]. This has triggered interest in international comparative trials with C-ions (Table 1) and investments in pre-clinical radiobiological research also in USA, resulting in recent P20 (https://grants.nih.gov/grants/guide/pa-files/PAR-13-371.html) and R01 (https://grants.nih.gov/grants/guide/rfa-files/RFA-CA-20-032.html) NCI grants on planning a C-ion facility and radiobiology research. 

With this increasing number of groups starting research in C-ion radiobiology, it is very important to assess where we are in our knowledge of C-ion radiobiology. The biological effects of charged particles depend on their charge and velocity, primarily on their LET. As shown in Figure 3, therapeutic beams of C-ions (typically 100–400 MeV/n) have LET ranging from 10 to 80 keV/μm, except for very high values in the distal edge. Therapeutic protons have LET ranging from 0.4 to 30 keV/μm, and their radiobiological properties have been recently reviewed in this journal [30]. It is well known that the peak of radiobiological effectiveness for several endpoints is around 100–200 keV/μm, typical values of α-particles [31], largely studied for protection [32] and therapy [33] with radionuclides, and for high-energy heavy ions, such as 1 GeV/n Fe-ions, whose radiobiology is largely studied for space radiation protection in the USA [34] and Europe [35]. Proton radiobiology is similar to X-rays, and indeed a constant RBE = 1.1 is used in particle therapy [36], even if the increased LET at the end of the range can have clinical consequences [37,38,39]. Alpha-particles are the most effective inducers of cell killing and late effects, and their biological effects have been studied for about a century [40,41]. Oxygen beams (Figure 3) can be more effective in the tumor region, but their higher LET in the normal tissue can lead to higher toxicity [42]. Carbon ions are therefore somewhere in between protons and α-particles, with the additional complication that nuclear fragmentation generates a mixed field, with only about 50% of the initial C-ions reaching the end of the range in a typical tumor treatment, the others producing lighter fragment particles [3,4]. 

In this paper, we will provide a review of the C-ion radiobiology studies in vitro and in vivo. We will focus on therapeutic beam energies, and cover both conventional radiobiology and molecular studies. We will always try to highlight how pre-clinical research can impact clinical trials in C-ion therapy.

## 2. DNA Damage

It is generally assumed that C-ions induce clustered DNA damage, difficult to repair, whereas the simple DNA double-strand breaks (DSBs) produced by electrons emitted by X-ray interaction in matter produce are simple and easy to repair (Figure 2). This statement is only partly true. A high fraction of clustered DSBs is measured after the exposure of mammalian cells to α-particles [43,44] or Fe-ions [45,46], but as noted previously, these particles have higher LET than therapeutic C-ions. The rejoining of DSB induced by C-ions at therapeutic energies is indeed quite efficient, as shown both by γH2AX (Figure 4) [47] and premature chromosome condensation (PCC) [48] assays. 

Nevertheless, an increase in the size of 53BP1 repair foci was observed in tumors coming from patients exposed to C-ions compared to X-rays [49]. High-resolution microscopy, such as single-molecule localization microscopy (nanoscopy, resolution ~10 nm), is necessary to carefully describe radiation-induced foci and their clusters [50]. Recently evidence of clusters of the DNA repair protein 53BP1 has been reported following exposure to ^15^N-ions (very similar to carbon ions), and these study also show that the number and relaxation of clusters is cell-type dependent [51]. Along with Monte Carlo models [52] and other in vitro cell data [53], these results suggest that therapeutic beams of carbon ions induce a higher fraction of clustered DSB than photons and protons, even if their distribution is much broader than after exposure to α-particles.

### 2.1. Repair Pathway Choice

The two main DNA DSB repair pathways are non-homologous end joining (NHEJ) and homologous recombination (HR), the latter only active in S and G2-phases of the cell cycle. It has been hypothesized that the production of DSB clusters by densely ionizing radiation triggers alternative, error-prone DNA damage repair (DDR) pathways [54,55,56]. A known alternative NHEJ pathway is based on DNA break resection followed by microhomology-mediated recombination [57]. The microhomology pathway is intrinsically error-prone and may lead to the formation of translocations [58,59]. Whilst resection is used in S/G2-phases as part of HR, it is never used in canonical G1-phase DDR pathway. It has been shown that resection in G1 increases with LET, but it is significantly high only at LET exceeding 100 keV/μm [60]. Therefore, a large fraction of DSB induced by C-ions is still processed by canonical NHEJ.

Some data show that mammalian cells resort more often to HR than NHEJ to process clustered DNA lesions [61]. An increased usage of HR has also been observed after exposure to therapeutic C-ions [62], but NHEJ remains the main pathway of choice, like for X-rays [63]. Surprisingly, measuring DSB repair kinetics in repair-deficient cell lines, some authors found that after proton therapy (low-LET, similar to X-rays), there was an almost complete shift from NHEJ to HR [64,65]. If little differences are observed between C-ions and X-rays in the DNA repair pathways, even smaller should be observed between protons and photons. The situation is further complicated by recent observations that the pathway repair choice may be dose-dependent, with more breaks being processed by NHEJ at high doses, when HR becomes saturated [66]. 

The repair pathway choice is an important point to clarify in particle therapy, because it directly affects the therapeutic strategy: should HR inhibitors be used in particle therapy, rather than NHEJ inhibitors? If alternative end-joining pathways are used after heavy ions, can we target these proteins to increase the effectiveness of C-ion therapy? We will see in Section 5.3 that combination of C-ions and targeted therapies is a hot topic in modern radiobiology.

### 2.2. Chromosome Aberrations

Incorrect DNA repair leads to chromosomal aberrations that are in fact commonly observed after exposure of human cells to carbon ions both in vitro and in vivo [67]. The RBE-LET curve is similar to that for cell killing, i.e., increase with LET and a peak around 100–200 keV/μm, followed by a decrease at higher values [68]. Chromosomal aberrations induced by heavy ions are also characterized by qualitative differences compared to X-ray induced aberrations, especially a strong increase in complex-type exchanges [69]. However, once again, these clear qualitative difference are only observed using α-particles [70] or Fe-ions [71], whose LET is substantially higher than therapeutic C-ions. Measurements of chromosome aberrations in peripheral blood lymphocytes of patients treated with X-rays and/or C-ions (Figure 5) did not show any qualitative difference in the aberration spectrum revealed by mFISH [72] or mBAND [73]. On the other hand, these experiments show fewer aberrations in C-ion treated patients due to the reduced volume of normal tissue exposed [74]. This effect can be significant for the immune response (see Section 5.4).

### 2.3. Summary

DNA damage and repair is the first step of the biochemical pathways leading to radiation-induced early and late effects, and it is at the base of the different biological response after low- and high-LET radiation. However, the rough distinction that X-rays produce “simple” DSB and C-ions “clustered” DSB is essentially incorrect. At therapeutic energies, C-ions have lower LET than α-particles, so the complexity at nm scale will be smaller. Many results obtained with monoenergetic, Bragg-peak C-ions do not reflect the real situation in the therapeutic scenario, or at least they only show what is experienced by a small minority of the tumor cells. The spatial distribution at μm scale remains quite different (tracks vs. uniform distribution) and this may also affect DSB misrejoining and result in the observed increase of chromosomal aberrations per unit dose. More experiments on DNA repair pathway choice are important to suggest therapeutic strategies in synthetic lethality scenarios.

## 3. RBE

The majority of early studies in C-ion radiobiology focused on the RBE for cell killing measured in mammalian cell lines. The RBE is a constant scaling factor (1.1) all along the SOBP in proton therapy, so it does not provide any benefit in terms of increased therapeutic window. However, in C-ion therapy, it is expected that the slow ions in the target will have higher RBE than the fast ions in the entrance channel (Figure 2). This means that the RBE increases the peak/plateau ratio compared to protons, and eventually widens the therapeutic window. 

This is nicely illustrated with the in vitro cell experiment in Figure 6. Chinese hamster ovary (CHO) cells in a water phantom were irradiated with two opposite beams of protons or C-ions. The calculation of the Local Effect Model (LEM) predicted the same survival in the entrance channel, while in the target, the higher RBE should have produced a lower survival in the cells exposed to C-ions. The results clearly show that the experimental data are close to the LEM predictions [75].

Figure 6 shows the potential advantages of the RBE for radiotherapy, especially for radioresistant (low α/β ratio) tumors. Yet it is clear that, because the RBE changes along the SOBP (Figure 3), a constant biological dose in the tumor requires a variable physical dose along the SOBP. The biological dose is actually the product of the absorbed dose (in Gy) with RBE, and was expressed in Gy(RBE) [76], but recently the ICRU has recommended the use of the same unit gray for the quantity indicated as RBE-weighted dose [77]. This is shown in Figure 7 [78] for carbon, helium, and hydrogen ions, even if at the moment no variable RBE is recommended by ICRU in proton therapy [79]. 

### 3.1. RBE Models

If we want a uniform biological effect (RBE-weighted dose) in the target region, we will have to calculate a variable physical dose (Figure 7), and this necessarily needs a mathematical model. Biophysical models are notoriously affected by large uncertainty, and efforts are ongoing to improve them. The uncertainty on the models is one of the reasons why proton therapy remains stuck to RBE = 1.1. The RBE is considered so small in proton therapy that resorting to mathematical models is perceived as an unnecessary complication. While the situation is currently changing also in proton therapy clinical settings [80], models have always been used in C-ion therapy, and they evolved during the past 20 years [81]. The Japanese centers started with the semi-empirical Kanai model [82] and switched more recently, with the introduction of beam scanning at NIRS, to the microdosimetric kinetic model (MKM) [83]. In Europe, an amorphous track structure model (LEM) was used since the GSI pilot project, and this model has evolved from the original LEM-I [84] to the latest version LEM-IV [75]. The differences between LEM-I and LEM-IV [85,86], LEM and Kanai model [87,88], and between the current models used in clinics and those under development [81,89] are quite significant. For radiation oncologists used to looking at the clinical response at the same dose, it is hard to compare the results obtained in Asia and Europe, and even the historical development of the data within the same institutes. This is deemed a serious caveat of C-ion therapy [24,90], and many radiation oncologists would like to have more precise models to reach a “universal” definition of RBE-weighted dose.

This desire is, however, not going to be fulfilled in the short term. RBE models use in vitro data, and this is already a major flaw, because radiosensitivity is affected by the tumor microenvironment and the general patient conditions. Moreover, the RBE depends on so many factors (dose, dose rate, intrinsic radiosensitivity, cell cycle phase, oxygen concentration etc.) that a large variance is unavoidable and will not be reduced by more experiments. 

### 3.2. In Vitro Studies

A comprehensive collection of in vitro RBE data for all ions and cell lines can be find in the PIDE database (https://www.gsi.de/bio-pide) [7]. In Figure 8, we reported all the RBE data relative to monoenergetic C-ion beams at different survival levels: initial slope (RBE_α_) used in radiation protection for estimating the radiation weighting factors; 50% survival (RBE_50_) corresponding to approximately 1 fraction of 2 Gy in conventional radiotherapy; and 10% survival (RBE_10_), commonly used as RBE reference in radiotherapy. Moreover, we divided the data in radioresistant (red; α/β < 4 Gy) and radiosensitive (blu; α/β > 4 Gy) cells. 

It is interesting to see that, within the variance, the shape of the RBE-LET remain the same, with values close to 1 in the low-LET region (corresponding to the EC in therapy) and a peak between 100–200 keV/μm, followed by a decrease (“overkill”). Clear quantitative differences are observed (note the different y-axis scale in the 3 plots). Moreover, the dependence from intrinsic radiosensitivity (α/β ratio) is very strong at low doses (RBEα), but practically disappears at higher doses (lower survival). This means that, at least for what the intrinsic, in vitro radiosensitivity is concerned, all tumors are expected to have RBE ranging between 2 and 3 after C-ion exposure (considering a range 40–80 keV/μm in the SOBP, neglecting the small edges).

The variance is certainly high, yet radiation oncologists use without hesitation the biological effective dose (BED) based on the Fowler formula [91] to calculate the expected effect of a fractionation change even if the α/β ratio in the BED formula is affected by a very large uncertainty and inter-individual variability–not differently from RBE. So, it is not clear why RBE uncertainty should be a showstopper for C-ion therapy, whilst α/β ratio uncertainty does not prevent clinical trials on hypo- or hyper-fractionation based on BED calculations. 

### 3.3. In Vivo Studies

Clinical centers starting with heavy ion therapy will necessarily begin with phase I/II safety and dose-escalation trials to find the optimal protocols. The question is, however, whether the accurate knowledge of the tumor RBE is really so important. RBE models are used to shape the physical dose to produce a uniform RBE-weighted dose, i.e., a constant effect (killing) in the tumor. Yet these days many radiotherapy protocols use highly non-uniform target doses, e.g., in stereotactic body radiotherapy (SBRT) [92] or in partial tumor irradiation protocols exploiting bystander effect [93] and targeting hypoxic regions (SBRT-PATHY) [94]. If we look at the simple experiment in Figure 6, it can be argued that the RBE in the normal tissue is what we really need to know, and the tumor RBE will provide an extra-benefit compared to protons. Normal tissue RBE should be better estimated with animal models, because the in vitro system is too simple to reproduce the late complications.

The experiment in Figure 6 uses the same cells in the entrance channel and SOBP, and the in vitro data in Figure 8 suggest that, at high doses, little differences in RBE will be observed between radiosensitive and radioresistant tissues. In the clinical scenarios, radiotherapy is limited by the late effects in normal tissue, which generally have low α/β ratio (~1–5 Gy), whereas tumors generally have high α/β ratio (~10 Gy) [95]. This justifies fractionation, which will spare normal tissues more than tumors. The RBE dependence for C-ions vs. the dose/fraction is shown in Figure 9 [96], where a constant β value is used [97] and RBE is given by an increase of the α parameter of the linear quadratic model.

Figure 9 shows that the RBE of C-ions can be actually higher in normal tissue (black curve) than tumor (red-blue) in conventional- or hyper-fractionation regimes, but the values become similar at high dose per fraction (hypofractionation), and even higher in the tumor at very high dose per fraction.

The RBE drop at high dose/fraction in Figure 9 has been confirmed by clinical data in lung cancer patients [98] and in several animal experiments on rat spinal cord [99,100,101,102,103,104], with a more significant fractionation effect in the entrance channel than in the SOBP, consistent with the LET dependence of the sparing effect (Figure 2). Acute skin reactions in mice-bearing tumors also support the RBE variation shown in Figure 9, i.e., higher RBE in tumor control than for skin reaction is observed in hypofractionation [105]. Table 2 shows experiments on normal tissue RBE or tumor growth delay after exposure of rodents to therapeutic C-ions. Only in a few cases normal tissue effects and tumor RBE was measured simultaneously [105,106]. Many experiments use single or split doses, and only a few afforded multiple fractions [102]. The experiments on a rat prostate model are the only one addressing tumor individual radiosensitivity, and nicely show that the large difference in X-ray sensitivity observed among three syngeneic prostate tumors is largely reduced using C-ions [107,108,109]. The reduced inter-patient variability is potentially a large advantage of C-ion therapy. 

Even if animal data are insufficient, the RBE values in Table 2 are actually in the range predicted from in vitro cell survival experiments (Figure 8), considering the doses used, and the dependence from the dose per fraction is consistent with the model in Figure 9. The RBE values in normal tissues are in the same range of those in the tumor tissue, but since normal tissue is exposed in plateau and tumor tissue in the SOBP, in the practical situation the RBE will be higher in the target as shown in Figure 6. 

An exception to this rule is the treatment of noncancer diseases, where actually the Bragg peak delivers high doses, generally in single fractions, to small normal tissue volumes [110]. This topic is now growing especially for non-invasive treatment of cardiac arrhythmias, already in clinical trials with SBRT [111,112]. Pre-clinical studies with C-ions therapy in a swine model gave excellent results [113,114]. Research on RBE of high does in cardiac sub-structures would be needed for optimizing the therapy.

### 3.4. Summary

The RBE of C-ions in mammalian cell cultures has been studied for many years and is very well known. Far less experiments have been performed with animal models, but they are utmost important to establish RBE in normal tissue, that eventually determine the tolerance doses [126]. Currently, mathematical models (LEM or MKM) based on the in vitro cell killing data are used to calculate a physical dose in the target able to provide a uniform effect in the whole tumor volume. However, this requirement of a uniform RBE-weighted dose in the tumor is not really justified, in view of the many protocols with SBRT giving high overdosage in the center or even providing partial tumor irradiation. On the other hand, the requirement of a uniform dose generates a non-uniform LET, and only a small fraction of the tumor is exposed to LET values above 100 keV/μm (Figure 3), comparable to α-particles. Are we losing the real advantages of heavy ion therapy in the attempt to reach a uniform (biological) dose distribution in the target? A recent retrospective analysis on the C-ion treatment plans at NIRS suggest that the LET is positively associated to local control of pancreas tumors. In particular, patients with higher minimum dose-averaged LET values in the gross tumor volume had lower probability of local failure compared to those with minimum LET values below 40 keV/μm [127]. Approaches to extend the high-LET regions (LET painting [128,129]), also using multiple ions [130], in the tumor are very promising for next-generation particle therapy, especially when specific radioresistant regions have to be targeted such as hypoxic volumes [131] or cancer stem cell niches [132]. This hypothesis can be tested first in animal models, and then in clinical trials.

## 4. Hypoxia

Hypoxia is one of the main features of solid tumors and is known to correlate with poor prognosis in cancer patients [133,134]. Hypoxia can be chronic or acute [135]. Chronic (diffusion-limited) hypoxia is caused by limitations in oxygen diffusion from tumor microvessels into the surrounding tissue. Acute (perfusion-limited) hypoxia is instead related to a temporary disturbance in perfusion, resulting in fluctuating microvascular oxygen supply. Solid tumors also contain regions of intermittent (cycling) hypoxia [136,137]. Even if chronic hypoxia a more typical conditions in solid cancers, acute or cyclic hypoxia can result in more aggressive phenotypes. 

In radiotherapy, the lack of oxygen directly affects the formation of ROS by indirect radiation effect, and therefore promotes radioresistance. The ratio of doses in hypoxic and oxic tissues (oxygen enhancement ratio, OER) can be as high as 3, making tumor control in hypoxia by radiation apparently impossible [138,139]. One of the main advantages of fractionation in radiotherapy is indeed reoxygenation, i.e., the possibility to supply oxygen to the surviving, previously hypoxic volumes, between fractions [140,141]. Hypoxia can also be tackled with specific drugs (hypoxia sensitizers), some of them currently in promising clinical trials [142,143,144,145]. Nevertheless, it remains a negative prognostic factor and a major hindrance to radiotherapy, especially in hypofractionation, when reoxygenation is limited or absent [146,147,148,149]. 

### 4.1. OER in Radiotherapy

OER has a complex dependence on the oxygen partial pressure (pO_2_) [150] and on the LET [151]. However, unlike the RBE, OER is poorly dependent on the dose [152]. A sharp distinction of normal tissue as 20% pO_2_ and tumor as 0% pO_2_ is unrealistic: normal tissues have a physoxia generally much lower than 20%, and tumors have variable oxygen levels around 0.3–4% [153]. The possibility of reducing the OER from ~3 to ~1 was the main motivations for the use of heavy ions in the LBL pilot trial [154]. Heavy ions have indeed predominantly direct effect, and are therefore less dependent on free radical production and oxygen concentrations. The clinical results in uterus cancer patients treated in Japan, where pO_2_ was measured in the individual patients and the outcomes compared, gave indications that hypoxia radioresistance can be reduced with C-ions [155]. Those results remain the only clinical evidence of effectiveness of C-ions in hypoxic tumors. However, the outstanding clinical results of C-ions in locally advanced pancreas cancer are suggestive of a reduced OER [156], because pancreas cancer is very hypoxic [157].

### 4.2. In Vitro Studies

A comprehensive in vitro study of the dependence of OER from pO_2_ and LET has been only recently published [158]. The results in Figure 10 show that the OER-LET trend is the same at all pO_2_ levels, but that the OER drops to 1 only at very high LET values. In tumor hypoxia, the OER is more likely to be 1.5–2, but is reduced to 1 only at LET > 200 keV/μm that are reached with C-ions only in the Bragg-peak. As also noted for RBE, the delivery of a uniform dose in the SOBP rather than uniform high-LET undermines the main advantages of heavy ion therapy, because the LET values are moderate rather than high. A more powerful strategy may be the use of heavier ions, in particular oxygen (Figure 11) [42,159,160,161], or a combination of multiple ions [130,131,162] to boost hypoxic regions visualized by PET or other molecular imaging [14,163,164,165]. 

In vitro cell experiments on radiation sensitivity have consistently shown that cells irradiated under acute hypoxia are more radioresistant than those exposed in chronic hypoxia [135]. However, after exposure to C-ions no significant differences are observed between acute and chronic hypoxia [166,167], and this can be a further clinical advantage. 

### 4.3. In Vivo Studies

As for RBE data (Table 2), animal experiments about OER in C-ion therapy are scanty. Hypoxia can be induced by clamping tumors implanted in mouse hind legs. With this technique, at NIRS it was reported a small decrease of the OER comparing X-rays (1.87) to C-ions (1.43–1.52 along the SOBP) [168]. Interestingly, tumors reoxygenated earlier after C-ions than X-rays [169], and the reoxygenation rate after C-ions in non-clamped tumors was similar to that in clamped tumors exposed to γ-rays [170]. These results suggest that C-ions may induce accelerated reoxygenation compared to photons. This is supported by more recent measurement of perfusion using dynamic contrast-enhanced magnetic resonance imaging (DCE-MRI) in rats exposed to single doses of C-ions and X-rays. C-ions caused larger changes in DCE-MRI parameters, suggesting faster perfusion and accelerated reoxygenation compared to X-rays [171]. 

Hypoxia-inducible factors (HIFs), transcriptional factors activated by hypoxia, are a known pharmacological target in cancer therapy [172,173]. In a xenograft model of human non-small cell lung cancer in BALB/c mice, irradiation with C-ions strongly reduced HIF-1α levels, a gene that can enhance survival by inducing hypoxia after exposure to X-rays [174]. Reduced HIF expression with C-ions compared to X-rays was also observed in cell experiments in vitro [175,176,177]. 

### 4.4. Summary

The possibility to crumble hypoxia radioresistance with carbon ions is very attractive, especially because heavy ion therapy is generally delivered in a few fractions, thus reducing the reoxygenation associated to fractionated regimes. Unfortunately, little clinical evidence and animal data are available to support this hypothesis, and more pre-clinical animal studies are certainly necessary. The results point to accelerated reoxygenation after C-ions. In vitro data are very extensive and complete, and suggest that C-ion alone may be insufficient to overcome hypoxia. Heavier ions or multi-ion treatments targeting hypoxic regions are very promising strategies, and radiobiological research in this direction is also necessary.

## 5. Molecular Radiobiology

Most of the radiobiology studies at large accelerators (LBL-Bevalac, NIRS-HIMAC, GSI-SIS18) focused on the “classical” topics discussed above, i.e., DNA repair, RBE and hypoxia. However, recently radiotherapy is moving toward precision medicine [178] and, consequently, pre-clinical radiobiology toward -omics and signaling pathways studies [179]. Here, we will review some “modern” topic in C-ion radiobiology, with a high growth potential in the coming years.

### 5.1. Radiogenomics 

The quest for biomarkers of tumor sensitivity is a “holy grail” of radiotherapy research, and has generally given disappointing results. Yet omics analysis in radiation oncology, unlike the studies in general oncology, has focused recently on normal tissue sensitivity with the formation of the Radiogenomics Consortium, including over 30 institutions worldwide [180,181]. The consortium has worked on patient’s samples using mostly single nucleotide polymorphism (SNP) with genome wide association studies (GWAS). Even if GWAS requires many thousands of patients to identify just modest percentages of a genotype associated to a given toxicity, the candidate-gene approach, e.g., using DNA repair genes, has given disappointing results [182]. Germline alterations in a single copy of a gene critical for radiation damage responses does not necessarily equate to increased risk of radiation-induced toxicity [183], with the possible exception of the second cancer risk in ATM heterozygous patients [184]. The Radiogenomics Consortium has been able to exclude a number of SNP previously candidate to be associated with normal tissue toxicity in radiotherapy [185], but has now identified a number of SNP associated with late toxicity in prostate and breast cancer patients [186,187,188,189,190]. The question is whether patients exposed to C-ions exhibit different biomarkers of normal tissue toxicity compared to conventional radiotherapy [191]. The Radiogenomics Consortium is currently analyzing prostate cancer patients treated at NIRS as well as proton therapy patients. Preliminary results, only presented in meetings [192], seem to show significant differences in the genes involved in toxicity induced by different radiation modalities. Results with the GWAS approach can significantly advance our possibilities to select patients for C-ion therapy, pending validation in prospective clinical studies of the GWAS results in radiotherapy, which is still missing.

Cell, tissue and animal studies of gene expression after exposure to C-ions have been recently reviewed in [193], so we refer to that paper for details. While radiation in general consistently activates genes involved in DNA repair, cell-cycle progression, and inflammatory pathways, there are many differences in the genes activated by X-rays and C-ions. Particularly interesting is the down-regulation by C-ions of genes involved in motility [194,195,196,197], which are generally upregulated by conventional radiotherapy [198]. This suggests that C-ion therapy may suppress angiogenesis and metastasis. The impact of these molecular signatures remains to be demonstrated in animal models and in the clinics. Taken together, transcriptomics seems to be able to help the selection of patients that will benefit from C-ion therapy, going beyond dosimetric analyses and models of normal tissue complication probabilities [199]. In addition to animal models, human organoids can also represent a useful tool to select differential panels of genes associated to exposure to X-rays and C-ions [200,201].

### 5.2. Cancer Stem Cells

Cancer stem cells (CSCs) are the main target of any therapeutic approach, being the only tumor cells with the potential of regenerate the tumor mass [202,203,204]. It is generally acknowledged that CSCs are radioresistant [205,206], based on their improved repair capacity [207], ROS scavenging [208] and their accumulation in hypoxic niches [209]. As we have already discussed above, C-ions DNA lesions are more difficult to repair, the direct effect prevails over indirect (ROS-mediated) effect, and the OER is reduced: clearly, C-ions seem to be the ideal tool against CSCs [132].

The increased effectiveness of C-ions against CSCs has been supported by several experiments in colon [210], pancreatic [211,212], breast [213], glioma [214,215,216] and laryngeal [217] CSCs. Even if the results are promising and fit the working hypothesis that C-ions are a powerful tool against CSCs [218], they cannot be considered conclusive. Most of the experiments are in vitro and exploit surface markers that are generally acknowledged as non-specific and also present in non-stem cells [219]. Better model systems and endpoints, including functional CSCs characterization, are needed to test whether C-ions can be a powerful tool against CSCs.

### 5.3. Combined Treatments

Radiotherapy is seldom administered alone. It is almost always part of a multi-modal treatment, that can include surgery, chemotherapy, targeted therapy, and immunotherapy. How can combined protocols be adapted to C-ion therapy? It is expected that more unexpected results will be found in combination experiment with C-ions than with protons, that are more similar to X-rays. Below, we will summarize results on the combination of C-ions with different drugs.

#### 5.3.1. Chemotherapy

Chemotherapy is typically associated with radiotherapy in many advanced solid tumors. Chemotherapy can be administered before (neo-adjuvant), during (concomitant) or after (adjuvant) the radiotherapy course. The benefits of radiochemotherapy on patients’ survival have been demonstrated in many cancers [220], but the optimal protocol depends on the biological sensitivity of the cancer cells. The “spatial cooperation” of radiotherapy and chemotherapy is used to target both the local and the systemic malignancy, and in this case the optimal way of combining these modalities is sequentially, thus reducing the overall toxicity of the combined treatment. However, very often, the advantage of radiochemotherapy is improved local control, which is based on independent cell killing, i.e., on the additional cell killing allowed by chemotherapy when the radiotherapy has to be stopped. Also, in this case, the application is generally in adjuvant settings. If instead the drug and radiation synergistically interact, then a concomitant setting has to be used. A typical example is the interaction of platinum compounds and X-rays in squamous cell carcinoma tumors of the head-and-neck region, where only the concomitant settings resulted in a significant survival benefit [221].

The effect of the combined treatment can be modelled extending the concept of BED to drug cell killing, and introducing ad-hoc coefficients to account for additive or synergistic interactions [222]. Using these, model, it could be shown that an additive function was able to describe the results with gemcitabine plus C-ions in pancreatic cancer in Japan [223]. The model parameters were derived from the data on conventional radiotherapy alone or in combination with chemotherapy, and extended to protons and C-ions by simply changing the dose BED corresponding to 50% overall survival (from 107 Gy for X-rays down to an RBE-weighted dose of 75 Gy with C-ions). 

In vitro experiments in human colon adenocarcinoma cells using different chemotherapeutic agents, including gemcitabine, showed a small S-phase specific sensitizing effect after X-rays [224], which is reduced with C-ions [225]. Results with pancreatic cancer cell lines also pointed to very small sensitizing effect of gemcitabine on C-ions lethal effect [226]. Similarly, purely additive effects where observed in glioblastoma cells exposed to C-ions in combination with temozolomide [227] or other platinum-based cytotoxic drugs [228] and in hepatocellular carcinoma cells exposed to C-ions in combination to gemcitabine [229].

#### 5.3.2. Targeted Therapy

Targeted therapy is the basis of precision medicine and consists in targeting specific pathways expressed by individual tumors [230,231]. Targeted therapy is typically delivered by monoclonal antibodies or small molecule drugs [232]. While conventional chemotherapy based on cytotoxic agents seems to have simply and additive effect to the cell killing induced by C-ions, “smart” targeted therapies can exploit specific properties of C-ions (e.g., reduced hypoxia and differential gene expression) for optimized therapeutic results. Typical targets of radioresistance are hypoxia-inducible factor 1 (HIF-1) and vascular endothelial growth factor (VEGF; involved in hypoxia; see Section 4); epidermal growth factor receptor (EGFR); PI3K/AKT/mTOR pathway; poly (ADP-ribose) polymerase (PARP); DNA-dependent protein kinase catalytic subunit (DNA-PKcs); heat shock protein 90 (Hsp90); and Hedgehog signaling pathway. Activation of these pathways in tumors is considered to be associated with radioresistance and poor prognosis in clinics. The combination of particle therapy and pharmacological targeting of these pathways has been recently reviewed in [233], so we refer to that excellent paper for details. In Table 3, we summarize the experiments involving C-ions. Clearly, only a few experiments have been performed, but the promising results justify further research on this crucial topic.

#### 5.3.3. Nanoparticles

Heavy metal nanoparticles have shown radiosensitizing properties in cells and animal models after irradiation with X-rays [246,247,248,249]. The mechanism is generally attributed to the production of low-energy electrons, especially Auger, by the interaction of photons with the high-Z metal atoms [250,251,252]. Even if enhancement factors for MV X-rays are generally modest (1.1–1.2), there are currently several attempts to bring nanoparticles in clinical radiation oncology settings. 

What about particle therapy? Because the physics of the interaction of charged particles is poorly dependent on the atomic number Z of the target, the sensitization is expected to be even smaller with particles than with photons. However, several experiments have shown significant sensitizations following proton irradiation, too (reviewed in [253,254]). Simulations for C-ions show large local dose enhancement in isolated nanoparticles, but do not identify any LET-dependence [255,256]. The additional problem in C-ion radiotherapy with nanoparticles is that the effect depends on the cross-section for the ion-nanoparticle interaction. With high-LET ions, the fluence per unit dose drops and the probability of direct interaction becomes poorly realistic at therapeutic doses.

#### 5.3.4. Immunotherapy

Harnessing the immune system with checkpoint inhibitors is nowadays considered the most promising strategy to defeat cancer, especially in metastatic patients [257]. 

Excellent results have been obtained with immune checkpoint inhibitors such as anti-CTLA4 and anti-PD1 antibodies in malignant melanoma [258]. However, severe immune-related side effects complicate the use of immunotherapy and limit its use in cancer patients [259]. Moreover, immunotherapy should be combined with local therapies to improve the survival in most solid cancers [260]. Because radiation has the potential to activate an anti-tumor immune response [261], it is an optimal candidate for combinations with immunotherapy [262,263,264,265,266,267,268,269,270,271]. Animal experiments have shown that the superior activity of radiation and dual immune checkpoint blockade is mediated by non-redundant immune mechanisms in cancer [272]. Following the initial positive results of this combination [273,274,275], hundreds of trials have been launched to test the safety and efficacy of radiation and immunotherapy in different tumors types, including lung cancer [276]. The PACIFIC trial has shown significant improvements in disease free survival [277] and overall survival [278] in stage III non-small-cell lung cancer (NSCLC) patients treated with Durvalumab after chemoradiotherapy. In a pilot/feasibility trial at Weill Cornell Medicine, a 30% disease control was achieved in stage IV NSCLC patients refractory to anti-CTLA-4 alone or in combination with chemotherapy by combining Ipilimumab with focal hypofractionated radiotherapy of a single metastasis [279]. Several strategies are under study to improve these results, including modifying the dose per fraction [280] and the number of metastasis irradiated [281]. Several strategies are under study to further improve these results, including modifying the dose per fraction [280] and the number of metastasis irradiated [281]. 

Considering the success and promise of the combination of X-rays with checkpoint inhibitors, the question is whether particle therapy can present additional advantages, and result in better outcomes [282]. This question is probably the most important for the future of particle therapy, considering the high cost compared to X-rays. It has been argued that particle therapy is advantageous compared to X-rays thanks to the physical properties, and in particular to the Bragg peak. In fact, much more normal tissue is spared using charged particles, and therefore more immune cells of the patient survive and can be exploited to enkindle a systemic response against the invasive malignancy [283]. This hypothesis is supported by the observation of reduced lymphopenia in esophageal cancer treated with protons [284] or C-ions [74] compared to conventional X-ray therapy. As lymphopenia is often a negative prognostic factor for cancer patients [285,286], this is an interesting working hypothesis that remains to be verified in clinical trials. 

Beyond the physical advantages, the question remains whether C-ions can potentiate immunotherapy more than protons or X-rays exploiting the unique high-LET biology described above. In this review we have described many times qualitative differences between C-ions and protons or X-rays effects, and more pronounced interaction with targeted therapy (Table 3). Since C-ions induce smaller DNA fragments that can more easily leak into the cytoplasm, it has been hypothesized that the innate immune pathway mediated by the cGAS-STING cytoplasmic DNA recognition [287,288,289] can be increased after exposure to C-ions [290]. The differences in DNA repair can be important for immunotherapy because PD-L1 expression in cancer cells is upregulated in response to DNA double-strand breakage, through the ATM/ATR/Chk1 kinase pathway [291]. Similar upregulation of PD-L1 has been recently shown in melanoma cells exposed to UV radiation [292]. The PD-L1 upregulation has also been recently shown in samples from patients treated with C-ions for uterine adenocarcinoma, compared to the expression before radiotherapy [293]. The activation of different DNA damage response pathways at high-LET, such as resection [60], may have different effects on the expression of immune receptors. It presently not known whether the radiation-induced upregulation of PD-L1 will actually translate into response to checkpoint inhibitors, but certainly this topic deserves a great attention [294]. 

Only a handful of animal experiments were carried out so far to compare C-ions and X-rays in combination with immunotherapy. The typical protocols of these experiments are shown in Figure 12. Generally, a target and an abscopal tumor are implanted in the hind limbs, only one is irradiated and different cocktails and timing of drug injection are tested. Endpoints include the response of the abscopal tumor and the growth of distal metastasis [295]. 

Pre-clinical studies of the type described in Figure 12 on the combination of immunotherapy with C-ions have shown promising results. First, an increased second tumor rejection was observed following injection of-pre-treated dendritic cells [296,297,298]. Second, reduced lung metastasis were measured after combination of C-ions with anti-CTLA4 and anti-PD-1 checkpoint inhibitors [299], and the effect was stronger when using C-ions than X-rays [300]. These results should be confirmed and can drive clinical trials.

### 5.4. Carcinogenesis

Cancer induction by C-ions has a dual interest. First, all heavy ion carcinogenesis is very important for space radiation protection [301], considering that late effects of radiation are now acknowledged as the main health risk for human exploration of the Solar system [302]. Second, it is important to assess the second cancer risk in C-ion therapy, a major concern especially for pediatric patients [303]. As already shown in Figure 3, C-ions have been selected among heavy ions exactly because they are expected to be nearly as toxic as X-rays or protons in the normal tissue, but more effective (high-LET) in the target region (Bragg peak). Nevertheless, the fear of second malignant neoplasms has prevented the use of C-ions for pediatric cancers, whereas protons, thanks to their ability to reduce the dose to the normal tissue, are considered the best option for treatment of solid malignancies in children [38,304]. It is important to point out that this prejudice against C-ion therapy in pediatric settings is not supported by data. Toxicity reported at HIT in the treatment of adolescents and children is similar for protons and C-ions [305]. Second malignancies in the large NIRS database of prostate cancer patients are actually lower in the cohort treated with C-ions than in patients treated with X-rays or surgery [306]. This is consistent with the rationale of the use of C-ions, and with the physical advantages of Bragg-peak therapy.

Animal carcinogenesis using C-ions at therapeutic energies has been recently reviewed and all details can be found in [307]. The RBE range approximately 1–10, depending on the type of irradiation (plateau or SOBP), dose, fractionation, age, and model system. The RBE values are generally lower than those measured using neutrons [308] or heavier ions, such as ^28^Si or ^56^Fe [309]. In general, very high RBE are only observed at very low dose. This is a problem for space radiation carcinogenesis risk, but not really for radiotherapy, because the low doses in the distal organs are caused by neutrons, not by charged particles [310]. The production of secondary neutrons in C-ion therapy using spot scanning is very low [303], actually so much lower than stray radiation in conventional radiotherapy that C-ions can be safely used to treat pregnant women [311]. The normal tissue in the entrance channel and in the margins of the planning target volume are exposed to moderate-high doses, where the RBE for carcinogenesis drops.

Due to the high uncertainty on the shape of the dose-response curve, it is hard to develop realistic mathematical models of radiation carcinogenesis. Comparative model studies of second cancer risk in patients exposed to C-ions and volumetric arc radiotherapy [312] or proton therapy [313] show only small predicted changes, that can change for different patients. 

### 5.5. Summary

Molecular radiobiology studies seem to open new horizons in C-ion therapy. Pre-clinical omics research suggests the activation of different genes, and much attention should now go to the analysis of C-ion patients by the Radiogenomics Consortium, that has produced panels of SNPs associated to normal tissue toxicity in patients treated with X-rays. Several experiments seem to show that C-ions can be very effective against CSCs, but these results beg for more precise methodologies in defining and identifying CSCs in vivo. Among the different combined modalities where C-ion can be used in clinical settings the most important is probably immunotherapy. In fact, immunotherapy is reporting exceptional results in combination with X-rays, but the physical and biological rationale suggest that C-ions can do better. Finally, the data on carcinogenesis show that, although C-ions are more effective than X.-rays in cancer induction, no significant differences are to be observed in clinical settings. 

## 6. Conclusions

The main motivation of carbon ion radiobiology has been pre-clinical studies for heavy ion therapy. Pre-clinical radiobiology in radiotherapy is now using molecular techniques and endpoints, biomarkers, and omics. For many years, C-ion pre-clinical studies studied steadfastly RBE and OER, the two classical motivations for heavy ion therapy (Figure 2). Almost half a century of cell and animal studies at LBL, GSI, and HIMAC (now backed by research in clinical centers such as HIT, CNAO, MedAustron, and SPHIC) gives us a deep knowledge of these two endpoints. It should be emphasized that this research has demonstrated that C-ion therapy is really “moderate” LET, not quite as high-LET as neutrons, α-particles, or Fe-ions, typically studied for radiation protection on earth or in space. In the tumor volume, C-ion therapy will unavoidably produce RBE<3 and OER>1. If we want to fully exploit the potential of high-LET radiation, as originally envisaged by Cornelius A. Tobias at LBL, we need to use either heavier ions (such as ^16^O; see Figure 11) or a different dose delivery, abandoning the restriction of a constant RBE-weighted dose and rather increasing the average or local LET values in the tumor. It is likely that these strategies will have a cost of a higher toxicity, and here more animal studies on normal tissue RBE (Table 2) and OER would be welcome.

In recent years, C-ion radiobiology is also moving toward molecular studies. This has led to a number of promising results. Even in proton therapy, molecular radiobiology shows differences compared to X-rays, probably linked to the increased LET of stopping particles. All these effects are more pronounced with C-ions. Pre-clinical studies suggest that C-ions can be exquisitely effective against CSCs, and can enhance the benefits of targeted therapy (Table 3) and immunotherapy. It is likely that combined treatments will be the mainstream topic in the coming years. The excellent clinical results of combined treatments using new drugs and conventional radiotherapy should alert the particle therapy community to explore this field more in detail. Preliminary results (see Section 5.3) are very encouraging, but they need confirmation in reliable pre-clinical models, either animal or human organoids. 

Carbon ion radiobiology has a high translational value, as it should lead to biologically driven treatment planning and biologically driven trials (see Table 1). The treatment planning for C-ions should include biological effects beyond RBE, such as OER and LET painting. The clinical trials should focus on patients that can have the maximum benefit from this therapy. In fact, the patient benefit with C-ions goes beyond the simple dosimetric advantage. Candidates for C-ion therapy include tumors with low α/β ratio, hypoxic tumors, tumors with high risk of metastasis, and those with a high number of CSCs. Inter-individual variability is expected to be reduced with C-ions, but this point also remains to be demonstrated in the clinics. More trials on benefit of combining targeted therapy and immunotherapy with C-ions are urgently necessary to assess the potential clinical benefit seen in pre-clinical research. Modern precision medicine stratifies the patients based on biomarkers. It will be very important to study the genetic markers of sensitivity in patients treated with C-ions in comparison to X-rays, a study ongoing within the Radiogenomics Consortium.

Finally, modern radiobiology is exploiting new technological improvements in radiotherapy, such as ultra-high dose rates (FLASH) [314,315,316] and spatially fractionated radiotherapy (minibeam) [317,318,319]. Research in this field needs high-intensity sources, such as synchrotron radiation, electrons, or protons (from cyclotrons), but the high intensities that can be achieved with the new heavy ion synchrotrons under construction [320] open the possibility of studying FLASH and minibeam with carbon or heavier ions. Some preliminary results with carbon ion interleaved minibeams in a rabbit brain [321] and calculations of molecular oxygen production under FLASH conditions [322] with C-ions suggest that these new methods can be highly beneficial in C-ion therapy. These new studies can pave the way of a future heavy ion therapy 10 years from now.

## Figures and Tables

**Figure 1 cancers-12-03022-f001:**
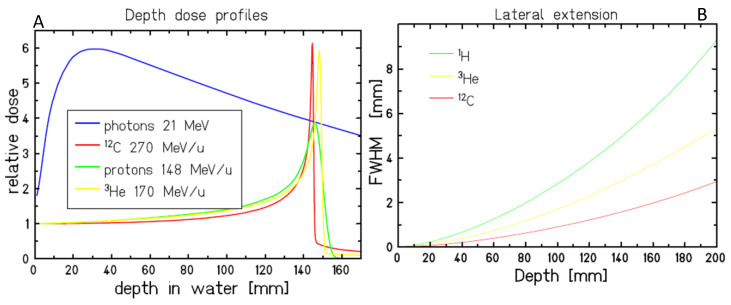
Physical properties of carbon ions in comparison to X-rays, protons, and helium ions. (**A**). Depth-dose distributions showing the Bragg peak for all ions at the same range, and the reduced straggling of heavier ions. (**B**). Lateral scattering is reduced by increasing the ion mass.

**Figure 2 cancers-12-03022-f002:**
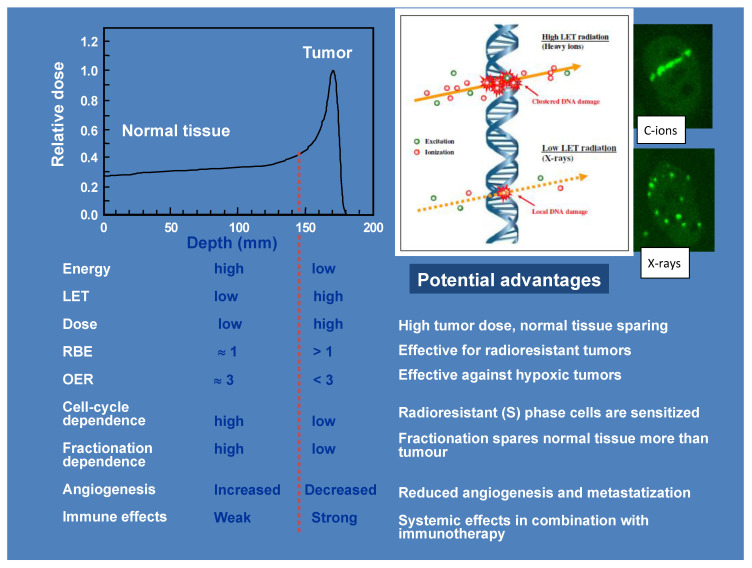
Summary of the physical and radiobiological properties of heavy ions along the Bragg curve. The figures on top right show a sketch of the quality of DNA damage and the corresponding γH2AX foci distribution with carbon ions and X-rays. Adapted from [12].

**Figure 3 cancers-12-03022-f003:**
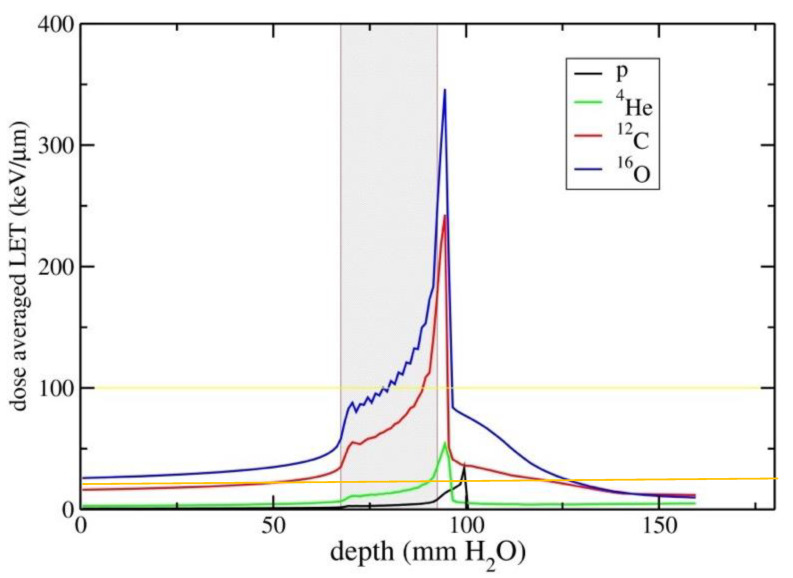
LET versus depth in tissue for a single SOBP of p, He, C, and O providing a uniform physical dose (2 Gy). The grey area represents the tumor region, a 2.5 × 2.5 × 2.5 cm^3^ volume centered at 8 cm in water. The yellow and orange lines are 100 and 20 keV/μm level, respectively. Figure from [15], distributed under Creative Commons CC-BY.

**Figure 4 cancers-12-03022-f004:**
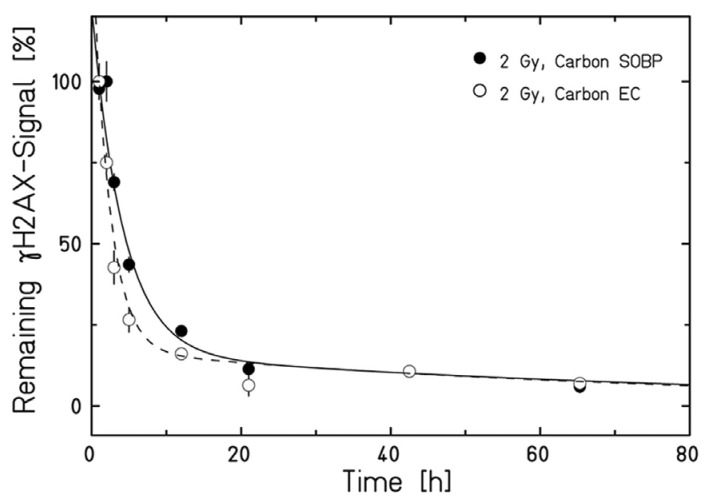
DNA DSB repair kinetics of human fibroblasts exposed in the entrance channel (EC) or SOBP (2.4 cm width at a water-equivalent depth of 16 cm, two opposite fields, 2 Gy dose). Figure from [47], distributed under Creative Commons CC-BY.

**Figure 5 cancers-12-03022-f005:**
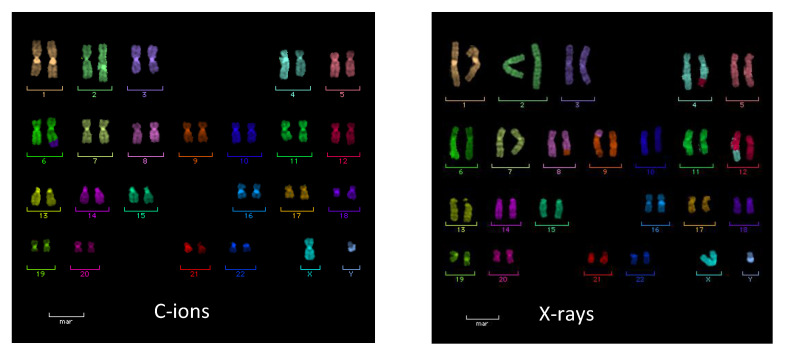
Karyotypes of peripheral blood lymphocytes from two esophageal cancer patients treated with either C-ions (RBE-weighted dose 36 Gy in 10 fractions, radiation field 61 cm^2^) or X-rays (41.4 Gy in 23 fractions, radiation field 141 cm^2^ ) at NIRS-QST, Japan. The C-ion (left) karyotype is 46, XY, t(6;18). The X-ray karyotype (right) is 46, XY, t (4;12), t(8;9). Samples were obtained as described in [74].

**Figure 6 cancers-12-03022-f006:**
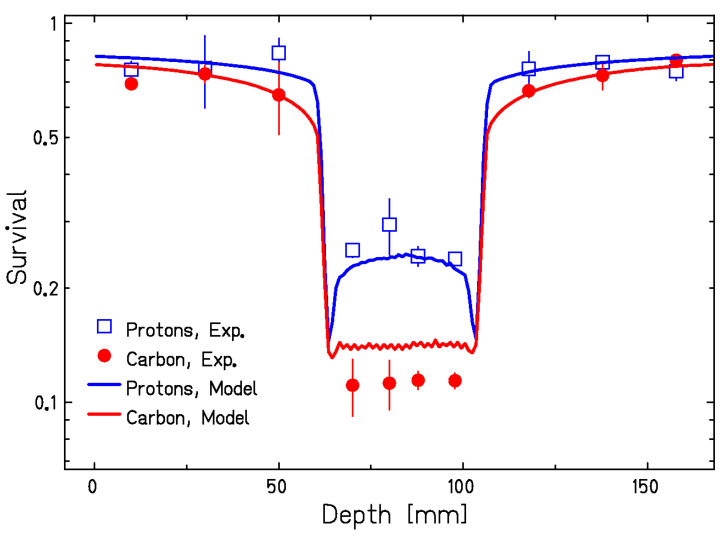
Survival of CHO cells irradiated in a phantom with two opposite beams of protons or C-ions. Proton and carbon ion irradiations were performed at the HIT facility using an active pencil beam scanning technique with energies of 90–120 MeV/u and 175–230 MeV/u for protons and carbon ions, respectively. The dose levels were 1.5 Gy in the entrance channel for both protons and carbon ions; in the center of target region, doses of 5.3 Gy and 3.9 Gy were applied with protons and carbon ions, respectively. Lines are predictions from the LEM-IV model. The model use the X-rays survival dose of CHO cells with α/β = 4 Gy and a threshold dose Dt = 20 Gy. Adapted from [75], reproduced with permission from Elsevier.

**Figure 7 cancers-12-03022-f007:**
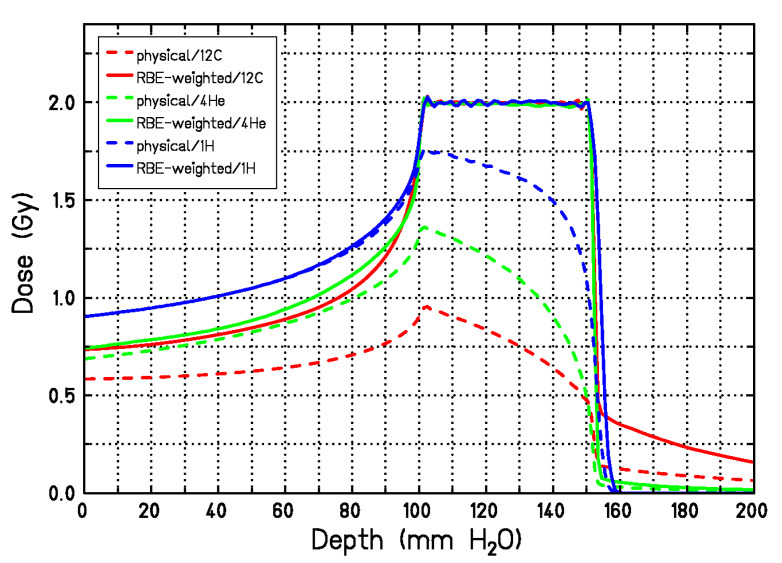
Physical (dashed lines) and RBE-weighted dose (solid lines) for a single 5-cm SOBP using protons, He- or C-ions. The physical dose shape is calculated with LEM-IV to achieve the same RBE-weighted dose in the target region for all ions. Figure modified from [78].

**Figure 8 cancers-12-03022-f008:**
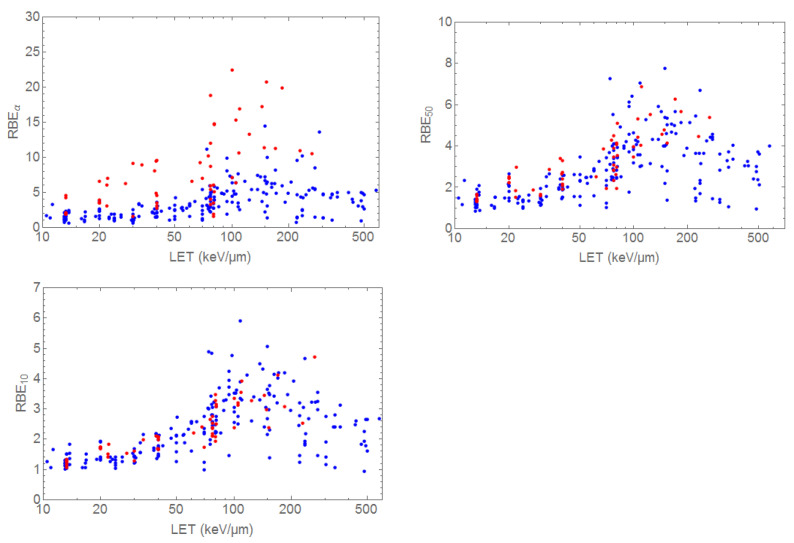
RBE of C-ions from the PIDE database. The 3 panels represent the RBEα, RBE_50_, and RBE_10_ for radioresistant (red; α/β < 4 Gy) and radiosensitive (blu; α/β > 4 Gy) cells. Plot courtesy of Dr. Thomas Friedrich, GSI.

**Figure 9 cancers-12-03022-f009:**
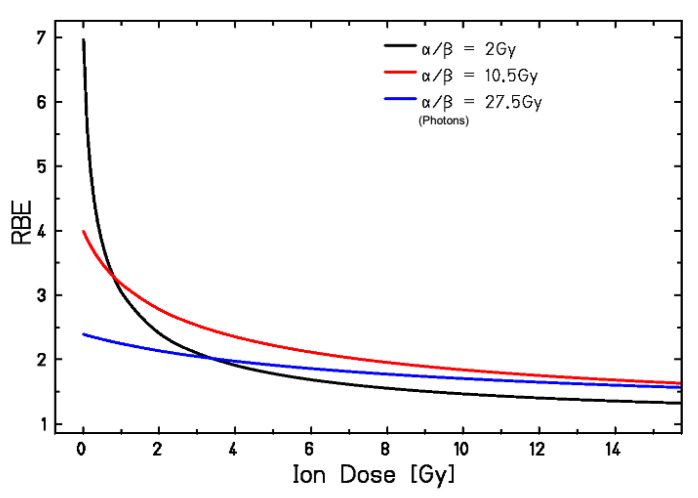
RBE of C-ions as a function of the dose per fraction calculated with the liner quadratic model for different α/β values. Figure from reference [96], reproduced with permission by Elsevier.

**Figure 10 cancers-12-03022-f010:**
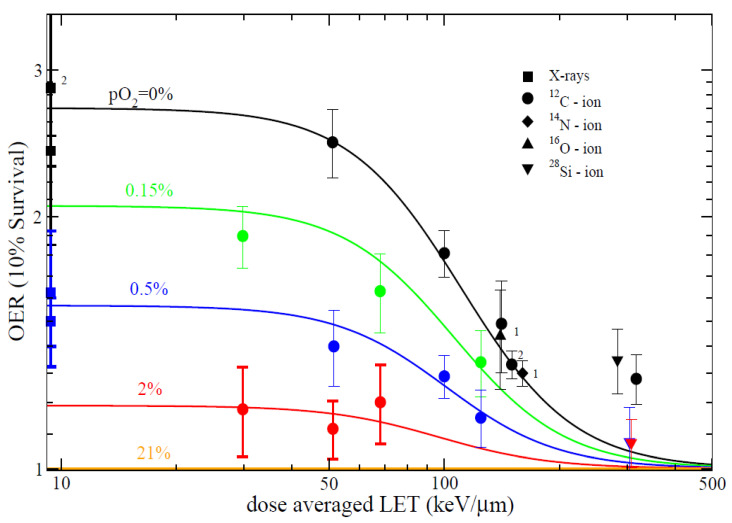
OER vs. LET of CHO cells at different oxygen concentration levels. Figure from reference [158], distributed under Creative Commons CC-BY.

**Figure 11 cancers-12-03022-f011:**
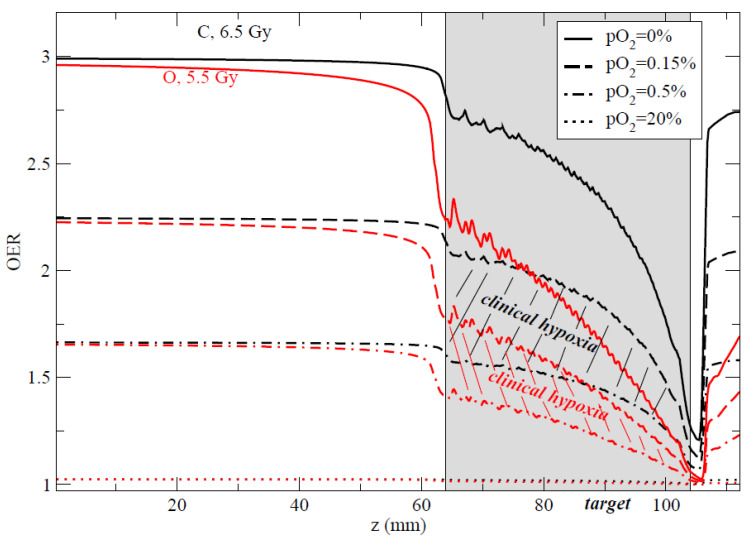
Comparison of the computed OER along an SOBP for carbon (black curves) and oxygen (red curves) at different pO_2_ levels. The hatched areas represent the clinical interesting regions for hypoxia (0.15% < pO_2_ < 0.5%). Doses indicated are prescribed RBE-weighted doses in the target to achieve iso-survival. Plans for using ^16^O in the clinics are currently under way at HIT (Heidelberg). Plot from [159], © Institute of Physics and Engineering in Medicine. Reproduced by permission of IOP Publishing. All rights reserved.

**Figure 12 cancers-12-03022-f012:**
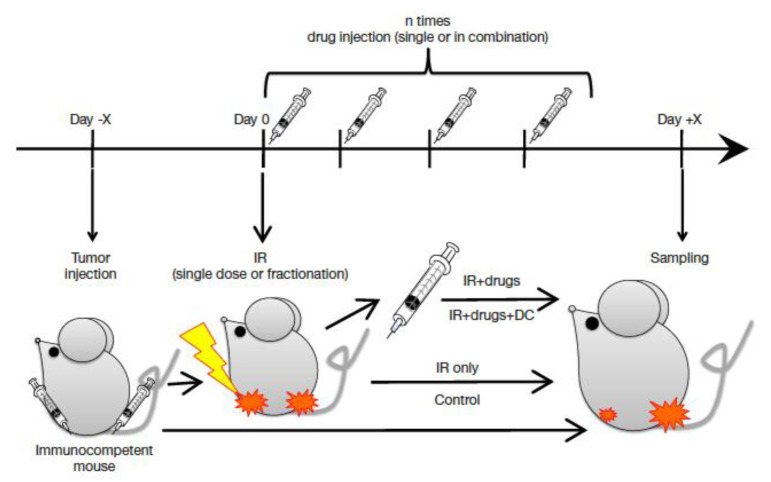
Typical protocol used in various laboratories to compare C-ions and X-rays in combination to immunotherapy. Reproduced from ref. [295], distributed under Creative Commons CC-BY.

**Table 1 cancers-12-03022-t001:** Ongoing randomized clinical trials comparing C-ions to either protons or photon therapy. Source: www.clinicaltrials.gov, updated October 2020.

Brief Title	ID	Sponsors	Phase	Condition	Arm 2
Trial of proton versus carbon ion in patients with chondrosarcoma	NCT01182753	Heidelberg University, Germany	III	Chondro-sarcoma of the skull base	Protons
Randomised trial of proton vs. carbon ion in patients with chordoma	NCT01182779	Heidelberg University, Germany	III	Chordoma of the skull base	Protons
C-ion radiotherapy for glioblastoma	NCT01165671 CLEOPATRA	Heidelberg University, Germany	II	Primary glio-blastoma	Protons *^,$^
Ion prostate irradiation	NCT01641185 IPI	Heidelberg University, Germany	II	Prostate cancer	Protons
Ion irradiation of sacrococcygeal chordoma	NCT01811394 ISAC	Heidelberg University, Germany	II	Sacrococcygeal chordoma	Protons
Neoadjuvant Irradiation of Retroperitoneal Soft Tissue Sarcoma	NCT04219202 Retro-Ion	Heidelberg University, Germany	II	Soft tissue sarcoma	Protons
Randomized C-ions vs. IMRT for radioresistant tumors	NCT02838602 ETOILE	Lyon University Hospitals, France	II	Adenoid cystic carcinoma and sarcomas	IMRT
Prospective trial comparing carbon ions to IMRT in pancreatic cancer	BAA-N01CM51007-51	NCI, USA	I/III	Locally advanced pancreatic cancer	X-rays *
Prospective multicenter randomized trial for pancreas cancer	NCT03536182 CIPHER	Toshiba and UT Southwestern, Dallas, TX	III	Locally advanced pancreatic cancer	X-rays *
Newly Diagnosed Glioblastoma	NCT04536649	Shanghai Proton and Heavy Ion Center	III	Primary glio-blastoma	X-rays *
Proton Versus Photon Radiotherapy for Nasopharyngeal Carcinoma	NCT04528394	Shanghai Proton and Heavy Ion Center	II	Naso-pharyngeal Carcinoma	Protons ^§^

* Combined with chemotherapy. ^$^ Boost following conventional chemoradiotherapy. ^§^ Both arms in combination with X-rays.

**Table 2 cancers-12-03022-t002:** Animal experiments measuring the RBE of C-ion beams in therapeutic conditions. Both normal tissue complication scores and tumor growth delay experiments are included.

Animal	Tissue	Doses (Gy)	Fractions	RBE	References
Rat	Spinal cord	15–50	1–18	1.5–3.5	[99,100,101,102,103,104]
Mouse	Skin reaction	10–80	1–6	1.2–3.0	[105,115,116]
		10–42	1	1.36	[106]
		10–60	1	1.3–2.16	[117]
		10–35	1	1.2–1.5	[118]
Mouse	Lung fibrosis	2–20	1–5	2.7–2.8	[119,120]
		10–42	1	1.51	[106]
Mouse	Mammary carcinoma	10–42	1	1.48	[106]
Mouse	Soft tissue sarcoma	10	1	3	[121]
Mouse	Fibrosarcoma	10–80	1–12	1.4–3.0	[105,122,123]
Nude mouse	Human esophageal cancer	2–20	1	2.02	[124]
Rat	Prostate carcinoma	10–60	1–2	1.6–2.3	[107,108,109,125]

**Table 3 cancers-12-03022-t003:** C-ions and targeted therapy combination experiments.

Target	Small Molecule Inhibitor	Doses (Gy)	Cells	Outcome	Ref.
EGFR	Cetuximab	1–4	Human laryngeal squamous cell carcinoma	Inhibition of invasion	[234]
mTOR	Temsirolimus	0.1–3	Hepatocellular carcinoma	Additive effects in cell killing	[229]
	Rapamycin	1–5	Chondrosarcoma cells	Sensitization of the C-ion effects	[235]
PARP1/2	Olaparib	1–5	Human pancreatic cancer cells	Sensitization of the C-ion effects	[236]
	PARP-1 knockdown	1–4	HeLa cells	Sensitization of the C-ion effects	[237]
	Talazoparib	2	Human glioblastoma stem-like cells	Sensitization of the C-ion effects	[238]
DNA-PKcs	Genistein	2–6	Human glioblastoma cell lines	Sensitization by inhibition of NHEJ	[239]
	NU7026	2	Human lung normal and cancer cells	Sensitization by inhibition of NHEJ	[240]
	NU7026	1–4	Hela cells, human breast cancer cells	Sensitization mediated by telomere-end capping	[241]
Hsp90	TAS-116	1–5	Hela, lung cancer and normal human fibroblasts, tumor xenografts	Radio-sensitization of both X-rays and C-ions	[242]
	PU-H71	1–7	HeLa derivative, human lung normal and cancer cell lines	Sensitization of cancer cell but not of normal cells	[243]
Hedgehog	GANT61	0.2–4	Prostate cancer cells Pediatric medulloblastoma	Sensitization and reduced migration	[244]
	GANT61	0.2–4	Human breast cancer cells	Reduced cell migration	[245]

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
