# Peer review of "Carbon Ion Radiobiology"

_cancers, 2020, doi:10.3390/cancers12103022_

Round 1

Reviewer 1 Report

The mansucript "Carbon ion radiobiology" by Tinganelli+Durante provides after a brief overview of the history of ion beams (also used as an argument for focussing on C-ions) a review on C-ion radiobiology studies in vitro and in vivo. The paper aims at showing on a scientific base the advantages of C-ion therapy.

71 in the description maybe "sketch" would be better than "cartoon"...maybe also a short sentence may be dropped on the shown differing distribution in the pictures right from the sketch

119 "as showed" please changed into "as shown"

for the arguments given in 125 to 129 you might consider new research on the nanostructure of damage and repair caused by heavy ions showing also new patterns changing as in

Bobkova et al. (2018) Recruitment of 53BP1 Proteins for DNA Repair and Persistence of Repair Clusters Differ for Cell Types as Detected by Single Molecule Localization Microscopy. International Journal of Molecular Sciences. 19. 3713. 10.3390/ijms19123713

and/or

Depes et al. (2018) Single-molecule localization microscopy as a promising tool for H2AX/53BP1 foci exploration. The European Physical Journal D volume 72, Article number: 158 (2018)

even though not everything is based on C but on very "close" ions as N

or as a reference for new parameters to look upon in research later in chapter5

158 may "followed by" be better than "following by"?

162 "also observed" or do you aim at saying there is no research on this with C ions?

166 "C ion"

in 164 it is written "X-rays or C-ions" the image given is Figure 5 "X-rays plus C-ions"...is there no image showing damage of X-rays and one for the damage of C-ions?

as the message is "no difference" between both, it is a bit confusing having an image of X plus C

188 "(+10%)" understanding problematic

330 good question

311 multiple ions OK...also multiple energies on these ions?

357 "of reducing the OER from ~3 to ~3" ?

399 "a few fractions"

424 it is mostly GWAS in literature...not GWA something

it is nice considering the problems of GWAS and writing honestly about an often ignored problem

529 maybe "generally" changed into " nowadays"

Table 2 is in this format somehow too long and is also on following page

625 "SNPs" not "SNP"...the usage of "elegant" in this context might be problematic for understanding

The paper is worth publishing as review, well written and understandable.

Author Response

See enclosed file

Reviewer 2 Report

In this manuscript by Tinganelli and Durante, the authors conducted a very extensive review of current knowledge on carbon ion radiobiology, and its potential clinical application. I have a few suggestions for the authors to make the contents more comprehensive.

  1. I suggested the authors to include a table comparing heavy ions and protons, which will provide a summary to readers regarding the similarities and differences in their radiobiologic properties.
  2. I also suggest the authors to include a table summarizing current “human” clinical studies of carbon ion radiation therapy (CIRT). This is to emphasize the clinical utility of CIRT.
  3. Figure 12 is not very informative in this manuscript, so can be removed.
  4. The authors can expand more on radiogenomics and precision medicine. For example, the authors can include the markers identified in GWAS, a little more details on genes potentially involved in carbon ion radiation mechanism of action, probability modeling including normal tissue complication probability.

Author Response

See enclosed file

Reviewer 3 Report

An excellent review comprehensively addressing all the relevant issues of carbon ion radiotherapy. I am happy to recommend it for publication without any changes.

Author Response

We thank the reviewer for her/his positive evaluation of our paper.

Reviewer 4 Report

In this review article, Tinganelli et al. describe radiobiologic studies governing C-ion radiotherapy. This is an important topic of an emerging treatment modality.

My main concern is whether the authors have permission to reproduce many of the figures. Some figures (such as Fig. 3, 6, etc.) are "adapted from" certain references. But a look at these references show almost exact replication. In addition fig. 12 is directly taken from the reference (#283).

The authors need to obtain rights to reproduce these figures or re-draw them by themselves in order to use them.

Author Response

Apologies for the confusion. We have now added the sources of the figures, the license, and for those figures not under a Creative Commons CC-BY license we have obtained permission from the publisher.

Reviewer 5 Report

General comments

This manuscript describes the review of biological studies in the carbon therapy. The developments of the biological studies in this field over last 10 years is remarkable. It is good time to review and publish the biological studies.

In particular, this manuscript focuses on the technologies that are useful for the development of the carbon beam therapy. It will be useful not only for researchers engaged in carbon therapy, but also for researchers engaged in the other radiation therapy.

Then, it should be published in "Cancers".

Minor comments

P11,L289   Figure 8 should be Figure 9.

P11,L301   and even higher in the tumor at very high dose per fraction

The horizontal axis in Fig.8 is LET and the vertical axis is RBE. From this figure, can it be said that RBE for normal tissue are similar to that for tumor and even larger in the high dose per fraction? may be Fig. 9.

Author Response

We thank the reviewer for her/his favorable review. We have corrected the text as suggested, wit the exception of L289, where we actually meant Figure 8, not Figure 9. On the other hand, we refer to Figure 9 in L301 now. Thank you.

Round 2

Reviewer 4 Report

The authors have addressed all major concerns. Well done!

Author Response

We thank the reviewer for her/his positive review. A few typos have been corrected or re-phrased.